# Preventive interventions for diabetic foot ulcer adopted in different healthcare settings: A scoping review protocol

**Açucena Leal de Araújo**[1]*, **Francisca Diana da Silva Negreiros**[2], **Raquel Sampaio Florêncio**[1], **Thiago dos Santos Garces**[3], **Virna Ribeiro Feitosa Cestari**[1], **Samuel Miranda Mattos**[3], **Samara Jesus Sena Marques**[1], **Francisca Eridan Fonteles Albuquerque**[4], **Wánderson Cássio Oliveira Araújo**[5], **Vanessa de Brito Poveda**[6,7], **Thereza Maria Magalhães Moreira**[1]

1 Graduate Program in Clinical Care, Nursing and Health, Ceará State University, Fortaleza, Ceará, Brazil, 2 Walter Cantídio University Hospital, Federal University of Ceará, Fortaleza, Ceará, Brazil, 3 Graduate Program in Public Health, Ceará State University, Fortaleza, Ceará, Brazil, 4 Professional Master's Program in Health Management, University of São Paulo, São Paulo, Brazil, 5 Health Sciences Librarian, Federal University of Ceará, Fortaleza, Ceará, Brazil, 6 School of Nursing, University of São Paulo, São Paulo, Brazil, 7 Brazilian Center for Evidence-Based Health: JBI Center of Excellence, University of São Paulo, São Paulo, São Paulo, Brazil

* a.leal09@hotmail.com

**Data Availability Statement:** No datasets were generated or analysed during the current study. All

## Abstract

### Background

Diabetic foot ulcers are challenging to heal, increase the risk of lower extremity amputation, and place a significant burden on patients, families, and healthcare systems. Prioritizing preventive interventions holds the promise of reducing patient suffering, lowering costs, and improving quality of life. This study describes a scoping review protocol that will be used to delineate the preventive interventions for diabetic foot ulcers employed in different healthcare settings.

### Methods

The scoping review methodology was formulated in accordance with the PRISMA extension guidelines for scoping reviews and informed by the procedural insights provided by the JBI methodology group. Studies with participants diagnosed with type 1 and type 2 diabetes, aged 18 years or older, without an active ulcer at baseline, and studies of preventive interventions for foot ulcers in various healthcare settings will be screened. The search strategy was developed in collaboration with a research librarian using the PRESS checklist and no time or language limitations were applied. Data will be analyzed and summarized descriptively, including characteristics of studies, participants, and interventions.

### Discussion

Understanding the strategies and gaps in diabetic foot ulcer prevention is critical. The literature can provide valuable insights for developing tailored interventions and strategies to

relevant data from this study will be made available upon study completion.

**Funding:** This study was partially funded by a grant to TMMM, from CNPq - Brazil, through INCT – CNPq nº 58/2022, Process number: 406540/2022-5.

**Competing interests:** The authors have declared that no competing interests exist.

effectively address these gaps, potentially accelerating progress toward improved outcomes in diabetic foot ulcer prevention.

## Review registration

Open Science Framework DOI 10.17605/OSF.IO/FRZ97 [June 19, 2023]

## Introduction

Diabetes is one of the fastest growing global health emergencies of the 21st century. Currently, there are approximately 537 million individuals worldwide living with diabetes mellitus (DM), with projections indicating a rise to 783 million by the year 2045 [1]. Lack of glycemic control, insulin resistance, microvascular and macrovascular dysfunction, among others, are possible complications [2]. Diabetic foot ulcers (DFUs) are the most common complication associated with neuropathy and/or peripheral arterial disease, causing high morbidity and mortality and significant financial costs to healthcare systems [3].

Diabetic peripheral neuropathy (DPN) reduces pain sensitivity, perception of plantar pressure, temperature and proprioception, and is present in half of all cases [4]. Symptoms of DPN can vary from person to person, but common complaints include numbness, tingling, burning, paresthesia, hyperesthesia and pain that begins in the fingers and soles and progresses over months or years to involve the entire foot and ankle [5].

Pain is the most important natural mechanism that alerts people to seek medical attention. Thus, people with DM may not notice their injuries until they get worse [6]. These symptoms tend to worsen at night and at rest, as movement relieves the pain associated with nerve fiber degradation, as poorly myelinated sprouts appear that are responsible for stimuli that cause painful sensation [7]. In this context, DPN has been confirmed as a relevant risk factor for the development of DFU, deformity and lower extremity amputation [8].

At least one-fifth of people with DM will develop DFU at some point in their lives. Its annual incidence is 2–5% and its prevalence is 4–10% [9]. Recurrences occur in 40% of cases within one year and in 65% of cases within three years [10]. Lower extremity ulcers account for one-fifth of hospitalizations and a lower extremity is amputated every 30 seconds as a result of the disease [9]. Its complications are responsible for 40–70% of all non-traumatic lower extremity amputations worldwide [11].

A systematic review of economic evaluations found that DFU is a source of significant patient suffering, but also a significant cost to the individual and the healthcare system. The cost burden of DFU requires from 6 days to 5.7 years of patient income to cover the cost of treatment, with variation depending on the treatment setting and strategy. The annual cost of treating DFU is significantly higher than the cost of treating non-diabetic foot ulcers, estimated at US$1.38 billion versus US$0.13 billion. DFU infection places an additional burden on healthcare systems. The cost per admission for patients with and without DFU infection was significantly higher in the former group ($11,290 versus $8,145) [12].

The current approaches used by health services to prevent and treat these injuries at an early stage are many and varied. These include: i) identification of the foot at risk; ii) regular inspection and examination of the foot at risk; iii) education of patients, families and healthcare professionals about diabetes and diabetic foot ulcers; iv) ensuring routine use of appropriate footwear; and v) treatment of risk factors for ulceration [13]. These strategies help to

minimize the incidence and recurrence of foot ulcers. However, efforts to prevent DFUs remain challenging and represent a significant global public health burden.

The risk factors for DFU may differ between countries and regions, reflecting a complex interplay of social, economic, cultural, and health determinants [14–16]. In areas with limited access to healthcare, such as parts of Africa and Asia, risk factors for DFUs may include loss of protective sensation, peripheral vascular disease, foot deformities, prior ulceration, and delayed initiation of healthcare follow-up [14, 16–19]. Additionally, individuals living with diabetes play an essential role in managing their own ulceration risk through foot self-care practices [20].

In regions with accessible healthcare systems, diabetic patients may have access to a wide range of services, including foot care education, regular foot screenings, follow-ups by multidisciplinary healthcare teams, and early treatment of precursor lesions to ulcers [21]. Conversely, in areas with underdeveloped or limited healthcare systems, access to these services may be restricted, resulting in a lack of follow-up and difficulty accessing preventive strategies for diabetic foot ulcers [22].

These differences should be considered when planning a scoping review. There may be variations in the care provided by each country's healthcare systems in managing diabetes and its complications. A study on challenges and innovative solutions to improve health outcomes in Latin America highlights the urgent need to prevent diabetes, as well as its related complications such as DFUs, due to the burden they pose on patients, societies, and healthcare systems. It presents the implementation of multidisciplinary and intersectoral strategies and collaborations as possible solutions to reduce the disease burden and improve health outcomes throughout Latin America [23].

A preliminary search for existing reviews on the topic was conducted and no reviews were identified. The databases searched were Medical Literature Analysis and Retrieval System Online (MEDLINE) via Pubmed, JBI Evidence Synthesis and Open Science Framework. Therefore, through this review, the aim is not only to understand the profile of patients with diabetes but also preventive interventions for DFUs, the professionals involved, the outcomes achieved through intervention implementation, healthcare settings worldwide, as well as weaknesses and strengths in the use of these preventive interventions. The contributions of this study lie in identifying best practices in DFU prevention in different healthcare contexts, providing robust evidence for patients, healthcare professionals, researchers, and policymakers.

## Objective

The scoping review objective is to delineate the preventive interventions for diabetic foot ulcers employed in different healthcare settings. Specifically, we aim to answer the following question: What is known about diabetic foot ulcer prevention interventions used in different healthcare settings?

## Methods

A scoping review protocol was developed using the procedural insights provided by the JBI Scoping Review Methodology Group [24]. We will follow the steps suggested by Arksey and O'Malley [25], with additional recommendations from Peters et al. [24]: 1) define and align the objective and research question; 2) develop and align inclusion criteria with the objective and question; 3) describe the planned approach to evidence search, selection, data extraction, and presentation of evidence; 4) search for evidence; 5) select evidence; 6) extract evidence; 7) analyze evidence; 8) present results; and 9) summarize the evidence in relation to the purpose

of the review, draw conclusions, and note any implications of the findings. The PRISMA extension for Scoping Reviews was used as the reporting framework [26].

The PPC (Population, Concept, Context) framework [24] was employed to formulate the research question. In this framework, P (population) represents individuals with either type 1 or type 2 diabetes mellitus, C (concept) pertains to preventive interventions for diabetic foot ulcers, and C (context) encompasses various healthcare scenarios.

## Inclusion criteria

We will include studies conducted with participants diagnosed with type 1 and type 2 diabetes mellitus, aged 18 years or older, and without an active ulcer at the start of the study. Pregnant women diagnosed with gestational diabetes will be excluded as hormonal changes occur during this period that may cause resistance to insulin action and are not a direct risk factor for DFU.

The main concepts of the study will include preventive interventions for diabetic foot ulcers related to foot self-management, footwear, structured education, foot-related exercises, orthotics, insoles [13], among others identified in the literature. Eligible studies must report outcome measures related to the use of interventions (preventive interventions that directly measure the presence or absence of DFU). For some definitions and examples of preventive measures, see Table 1.

Regarding the context, we will include studies of preventive interventions used in different healthcare settings: primary, secondary, and tertiary healthcare facilities; home and community settings; and private clinics.

## Types of sources

It will consider quantitative and qualitative primary studies that address issues related to interventions for the prevention of diabetic foot ulcers, without time or language limitations.

**Table 1. Chart 1.** Interventions for the prevention of diabetic foot ulcers.

| Interventions | Definitions | Examples |
|---|---|---|
| Feet self care | Feet self-care interventions that the patient can perform at home. | Inspecting the feet, washing the feet, carefully drying between the toes, cutting the nails, using moisturizers to hydrate the skin, etc. |
| Feet self management | Advanced assistive interventions (supportive technology) that the patient can use at home. | Home monitoring systems (infrared thermometer, thermal scales), lifestyle interventions, telemedicine/telehealth, technological applications, peer support programs. |
| Footwear | Broadly defined as any equipment for use on the feet, which may include insoles. | Prefabricated therapeutic footwear, custom-made therapeutic footwear. |
| Structured education | Any education modality offered in a structured manner. It can take various forms. | Educational programs, individual verbal education, motivational interviews, group educational sessions, video education, booklets, software, questionnaires, and pictorial education through cartoons or descriptive images. |
| Foot exercises | Any physical exercise specifically targeting the foot or lower limbs with the aim of altering foot physiology. These exercises are provided and/or supervised by a physiotherapist or healthcare professionals with appropriate training. | These exercises may include stretching and strengthening of the foot and ankle muscles and functional exercises such as balance and gait training. |
| Toe orthosis | Promotes alignment, pressure relief, spacing, and protection in the toes. | Custom-made silicone orthosis. |
| Insoles | May be a custom-made insole for the individual's foot using foot impression, mold, phenolic foam, and/or plaster, providing redistribution/relief of plantar pressure, cushioning, and postural alignment, or a flat or contoured "ready-to-use" insole made without reference to the patient's foot shape. | Custom-made insoles, prefabricated insoles. |

**Source:** Adapted from the Practical Guidelines of the International Working Group on the Diabetic Foot [13]

Opinion articles, experience reports, ongoing trials, incomplete articles, book chapters, editorials and proceedings will be excluded.

## Search strategy

The following databases will be searched: Scopus, Web of Science, MEDLINE via PubMed, Embase, Cummulative Index to Nursing and Allied Health Literature (CINAHL), The Cochrane Central Register of Controlled Trials (Cochrane Library), and Latin American and Caribbean Health Sciences Literature (LILACS) via Virtual Health Library (VHL).

To reduce publication bias and identify as much relevant evidence as possible, grey literature sources will also be consulted: Google Scholar, Brazilian Digital Library of Theses and Dissertations (BDTD in Portuguese), Catalog of Theses & Dissertations (CTD) of the Coordination for the Improvement of Higher Education Personnel (CAPES in Portuguese), Open Grey, and Open Access Theses and Dissertations (OATD). The first 100 records retrieved will be ranked by relevance.

Based on the established research question, we developed a search strategy using the steps of the Extraction, Conversion, Combination and Use (ECCU) method proposed by Araújo [27] (S1 and S2 Appendices). To ensure the feasibility of the method, we followed the Peer Checklist Review of Electronic Search Strategies (PRESS) guidelines [28].

Four controlled vocabularies were used to develop the search strategy: Medical Subject Headings (MeSH), EMTREE from Embase, CINAHL Subject Headings, and Health Science Descriptors (DeCS in Portuguese). These controlled terms were combined with free natural language terms using the Boolean operators AND and OR to increase the sensitivity of the search and to obtain a broader range of relevant results [29].

To maintain an appropriate level of specificity, we chose to use terms related to the diabetic foot rather than specific terms related to diabetes mellitus and diabetic patients. Although the research question is focused on prevention and the population of interest is logically made up of people who have not yet developed diabetic foot, this logic would not directly apply to the search strategy. This is because the search strategy establishes a link between prevention measures and diabetic foot. If we included the broader terms (diabetes mellitus and diabetic patients), specificity would be compromised and work outside the scope of the research would be retrieved.

To broaden the scope of our search results, we interpreted the term associated with the concept including terms such as primary prevention and prophylaxis, which are directly related to disease prevention. To prevent the loss of valuable documents due to contextual limitations, we opted against explicitly defining terms from this thematic block in our search strategy. Instead, document selection will be based on thorough reading, as suggested by the authors, aiming to enhance qualitative control over the results obtained from bibliographic databases.

We will create a Portuguese variant for the LILACS bibliographic database. For the Scopus and Web of Science databases, which do not have an integrated controlled vocabulary, we will use the standard search strategy. For other bibliographic databases, we will use the standard search strategy with their respective subject headings.

All information sources, databases, and grey literature will be searched on the same day to mitigate potential bias. Initially, a preliminary search of the MEDLINE database via PubMed was conducted on November 28, 2023 (S3 Appendix), and it will subsequently be adapted for use with other electronic databases.

Ultimately, the literature search will conclude with a thorough review of the reference lists in the included documents, supplemented by insights and recommendations from experts in diabetes and diabetic foot ulcers.

## Study/source of evidence selection

The results obtained from information sources will be exported to the Rayyan QCRI Reference Manager [30]. This tool allows for fast and accurate processes, as it allows for the removal of duplicate studies, selection, and screening of studies. In addition, it maintains methodological rigor and transparency among reviewers, as it allows for blind review, thus avoiding potential bias [30].

Two investigators will independently review the titles and abstracts to identify potential articles of interest, and in the next step, the full text of preselected articles will be reviewed to confirm inclusion. A third investigator will resolve conflicts in the selection process.

The Cohen's kappa coefficient will be computed to assess agreement between reviewers during various selection phases, employing the following classification: 0–0.20 (none), 0.21–0.39 (minimal), 0.40–0.59 (weak), 0.60–0.79 (moderate), 0.80–0.90 (strong), and above 0.90 (almost perfect). If the agreement is 0.79 or less, training sessions will be organized among the raters to enhance the reliability of the process. To facilitate this, a subset of 30 studies will be selected for testing with the raters.

The studies that meet the inclusion criteria will proceed to the second phase. This includes full-text reading of the studies by the same reviewers independently to confirm eligibility and selection of studies for inclusion in the review. Any inconsistencies will be discussed with an additional reviewer. Finally, the third stage involves manual searching for references of included studies. The entire process of identification, screening, and inclusion will be documented in the PRISMA flowchart.

## Data extraction

Data extraction will be performed by two independent reviewers using a Microsoft Excel® spreadsheet. The accuracy of the information will be checked by the third reviewer. Disagreements and uncertainties will be discussed until a consensus is reached among all authors. Information mapping will be based on the adaptation of the JBI instrument to characterize scientific production [24].

The following data will be extracted from the studies: i) characteristics of the study (author, country of origin, year of publication, journal, objective(s), and study design); ii) characteristics of the participants (sample size, sex, average age, and risk of ulceration); iii) characteristics of the intervention (type of intervention, sample recruitment site, duration of intervention, professionals involved, and main outcomes of preventive interventions); and iv) context (healthcare settings, S3 Appendix). Additional important variables will be extracted during the full-text reading process.

The draft version of the data extraction tool may be modified and revised as necessary during the data extraction process, with any adjustments documented in the scoping review report. Additionally, to address selective reporting within studies, a proactive approach will be taken to identify unpublished studies. Key researchers in the field will be contacted to inquire about any relevant unpublished data. Furthermore, in cases where data are incomplete or further clarification is needed, authors will be contacted directly to request missing or additional information.

## Data analysis and presentation

The results will be analyzed and synthesized descriptively, employing a narrative synthesis approach. Initially, the studies will be grouped to offer a comprehensive overview of the scope, nature, and distribution of those included in the review, presented in a characterization table.

Subsequently, the results will be discussed using thematic categories identified by the authors after thorough reading and exploration of the research encompassed in the review.

## Discussion

DFUs contribute to the burden of patients, societies, and healthcare systems. For patients, foot ulcers can cause pain, decreased mobility, and increased risk of infections, resulting in serious complications such as amputation. They also impose economic burdens on societies due to treatment costs, including medical expenses, loss of productivity, and the need for long-term care. Healthcare systems face increasing demands for resources and services to treat foot ulcers, such as hospitalizations, surgeries, and specialized wound care [31].

In response to the need to prevent DFUs, countries around the world have implemented a variety of initiatives. These include foot care education programs, footwear and insole provision, and the use of technology and innovation to improve disease management [32]. The integration of these strategies aims not only to mitigate the risk factors associated with ulceration but also to empower individuals and communities to adopt healthy behaviors in their daily lives [33].

This scoping review protocol endeavors to make a significant contribution to the field of diabetic foot ulcer prevention. The meticulous design of the search strategy ensures the comprehensive retrieval of relevant evidence. By consulting various databases and grey literature sources, utilizing controlled vocabularies, and applying Boolean operators, the protocol maximizes sensitivity in identifying pertinent studies. Through this comprehensive approach, the review aims to gather valuable evidence that can inform future research directions and enhance understanding of effective interventions for diabetic foot ulcers.

Expected outcomes of this thorough review process include the identification of research gaps and the synthesis of evidence to guide the development of tailored interventions. Ultimately, these contributions are anticipated to lead to improved patient outcomes and advance healthcare practices in the prevention of diabetic foot ulcers.

## Supporting information

**S1 Checklist. PRISMA-P (Preferred Reporting Items for Systematic review and Meta-Analysis Protocols) 2015 checklist: Recommended items to address in a systematic review protocol\*.**
(DOC)

**S1 Appendix. Review question and search strategy formulated for the MEDLINE/PubMed database. Source:** Adapted from Araújo [27].
(DOCX)

**S2 Appendix. Search strategy.** Database and name of platform used: MEDLINE/PubMed. Search conducted: November, 2023.
(DOCX)

**S3 Appendix. Data extraction tool.** DM: diabetes mellitus. \* Other variables considered important may be extracted during the full-text reading.
(DOCX)

## Author Contributions

**Conceptualization:** Açucena Leal de Araújo, Francisca Diana da Silva Negreiros, Raquel Sampaio Florêncio, Samuel Miranda Mattos, Thereza Maria Magalhães Moreira.

**Data curation:** Açucena  Leal de Araújo, Francisca Diana da Silva Negreiros, Raquel Sampaio Florêncio, Thiago dos Santos Garces, Wánderson Cássio Oliveira Araújo.

**Formal analysis:** Açucena  Leal de Araújo, Francisca Diana da Silva Negreiros, Thiago dos Santos Garces, Samuel Miranda Mattos, Samara Jesus Sena Marques, Francisca Eridan Fonteles Albuquerque.

**Investigation:** Açucena  Leal de Araújo, Francisca Diana da Silva Negreiros, Raquel Sampaio Florêncio, Thiago dos Santos Garces, Samuel Miranda Mattos.

**Methodology:** Açucena  Leal de Araújo, Francisca Diana da Silva Negreiros, Raquel Sampaio Florêncio, Thiago dos Santos Garces, Samuel Miranda Mattos.

**Project administration:** Açucena  Leal de Araújo, Thereza Maria Magalhães Moreira.

**Supervision:** Açucena  Leal de Araújo, Vanessa de Brito Poveda, Thereza Maria Magalhães Moreira.

**Validation:** Virna Ribeiro Feitosa Cestari, Vanessa de Brito Poveda.

**Visualization:** Virna Ribeiro Feitosa Cestari, Vanessa de Brito Poveda.

**Writing – original draft:** Açucena  Leal de Araújo, Francisca Diana da Silva Negreiros, Raquel Sampaio Florêncio, Thiago dos Santos Garces, Virna Ribeiro Feitosa Cestari, Samuel Miranda Mattos, Samara Jesus Sena Marques, Francisca Eridan Fonteles Albuquerque, Wánderson Cássio Oliveira Araújo, Vanessa de Brito Poveda, Thereza Maria Magalhães Moreira.

**Writing – review & editing:** Açucena  Leal de Araújo, Francisca Diana da Silva Negreiros, Raquel Sampaio Florêncio, Thiago dos Santos Garces, Virna Ribeiro Feitosa Cestari, Samuel Miranda Mattos, Samara Jesus Sena Marques, Francisca Eridan Fonteles Albuquerque, Wánderson Cássio Oliveira Araújo, Vanessa de Brito Poveda, Thereza Maria Magalhães Moreira.

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
