## [Decision Letter · Decision Letter 0]

25 Mar 2024

PONE-D-24-03451Preventive interventions for diabetic foot ulcer adopted in different healthcare settings: A scoping review protocolPLOS ONE

Dear Dr. Leal de Araújo,

Thank you for submitting your manuscript to PLOS ONE. After careful consideration, we feel that it has merit but does not fully meet PLOS ONE’s publication criteria as it currently stands. Therefore, we invite you to submit a revised version of the manuscript that addresses the points raised during the review process.

We look forward to receiving your revised manuscript.

Kind regards,

Palash Chandra Banik, MPhil

Academic Editor

PLOS ONE

Journal Requirements:

"This study was partially funded by the Coordination for the Improvement of Higher Education Personnel – CAPES/Brazil - Financial Code 001."

 "This study was partially funded by the Coordination for the Improvement of Higher Education Personnel – CAPES/Brazil - Financial Code 001."

Reviewers' comments:

Reviewer's Responses to Questions

**Comments to the Author**

1. Does the manuscript provide a valid rationale for the proposed study, with clearly identified and justified research questions?

Reviewer #1: Partly

Reviewer #2: Yes

Reviewer #3: Yes

2. Is the protocol technically sound and planned in a manner that will lead to a meaningful outcome and allow testing the stated hypotheses?

Reviewer #1: Partly

Reviewer #2: Yes

Reviewer #3: Yes

3. Is the methodology feasible and described in sufficient detail to allow the work to be replicable?

Reviewer #1: Yes

Reviewer #2: Yes

Reviewer #3: Yes

4. Have the authors described where all data underlying the findings will be made available when the study is complete?

Reviewer #1: Yes

Reviewer #2: Yes

Reviewer #3: Yes

5. Is the manuscript presented in an intelligible fashion and written in standard English?

Reviewer #1: Yes

Reviewer #2: Yes

Reviewer #3: Yes

6. Review Comments to the Author

You may also provide optional suggestions and comments to authors that they might find helpful in planning their study.

Reviewer #1: Background:

The introduction provides a comprehensive overview of the differences in diabetes and foot care across countries, emphasizing the need to consider these variations in a scoping review. The mention of diverse risk factors for Diabetic Foot Ulcers (DFUs) and the impact of healthcare system differences adds depth to the understanding of regional disparities.

Comments and Questions:

The statement rightly underscores the importance of recognizing and addressing variations in diabetes and foot care during a scoping review. However, specific examples of how risk factors for DFUs vary between countries or regions would enhance the clarity of the argument.

Exploring how healthcare system differences impact diabetes management, especially in foot care, would provide valuable insights. More elaboration on this aspect could strengthen the connection between the review and healthcare system contexts.

Further details on the findings of the study in Latin America, including challenges and innovative solutions, would be beneficial. Understanding the applicability of these insights to other regions could enhance the relevance of the scoping review.

Clarifying how DFUs contribute to the burden on patients, societies, and health systems, and providing examples of variations across countries, would add depth to the discussion.

Mentioning existing initiatives or strategies in certain countries to address the urgent need for preventing diabetes and its complications, as highlighted in the Latin American study, would contribute to a more comprehensive understanding.

According to:

The emphasis on understanding global preventive measures for DFUs is well-stated, highlighting the importance of planning targeted health interventions and monitoring progress in health system performance.

Comment:

Clarifying why a scoping review is necessary and detailing the expected contributions to new knowledge would strengthen this section. Further clarification on the specific information sought through the scoping review would be beneficial.

Objective:

The suggested clarification of the scoping review's aim, focusing on preventive interventions, patient consent, and contextual factors, is well-articulated and aligns with the broader goals outlined in the background.

Methodology:

Referencing the methodology paper on scoping reviews is relevant, providing a basis for best practice guidance. The inclusion of the full citation enhances the credibility of the review.

Chart 1:

The comment about the difficulty in understanding Chart 1 raises a valid concern. Additional clarification on how it differs from the objective or providing an example could improve comprehension.

Data Extraction:

Including an example of a data extraction sheet along with contextual information is an excellent suggestion. It would enhance transparency and help readers better understand the process.

In summary, the review is well-structured and informative, with the suggested improvements aimed at enhancing clarity and depth in specific areas.

Reviewer #2: It is a very well written manuscript, well referenced (except for reference#11), and adequate protocol for a scoping review. However, I find that the publication of a scoping review protocol does not contribute to scientific knowledge very much. This type of publication does not provide more information than what is already registered and made available to all (through OSF in this case). It feels as if the authors are duplicating publications from a single project which is borderline unethical.

Reviewer #3: This scoping review is interesting since it will map the preventive interventions available for diabetic foot ulcers adopted in different healthcare settings.

Introduction

Several systematic reviews (including a meta-analysis) related to the prevention of foot ulcers in patients with diabetes are available, such as:

- Prevention of foot ulcers in the at-risk patient with diabetes: a systematic review

- Prevention, assessment, diagnosis and management of diabetic foot based on clinical practice guidelines: A systematic review

- Preventing foot ulceration in diabetes: systematic review and meta-analyses of RCT data

Therefore, please elaborate more on the novelty of this scoping review compared to existing reviews.

Methods:

How will the authors conduct "critical appraisal" for the included studies?

7. PLOS authors have the option to publish the peer review history of their article (what does this mean?). If published, this will include your full peer review and any attached files.

Reviewer #1: No

Reviewer #2: No

Reviewer #3: **Yes: **M. Rifqi Rokhman

---

## [Author Response · Author response to Decision Letter 0]

15 Apr 2024

Dear Editor and Reviewers of Plos One,

We sincerely appreciate your insightful feedback on our manuscript. We have carefully considered each of your comments and suggestions and would like to address them accordingly. Please refer to the document attached in the "attach files" step, named as Response to Reviewers. The document does not contain any identifying information.

Sincerely,

The authors.

---

## [Decision Letter · Decision Letter 1]

19 Jun 2024

Preventive interventions for diabetic foot ulcer adopted in different healthcare settings: A scoping review protocol

PONE-D-24-03451R1

Dear Dr. Araújo,

We’re pleased to inform you that your manuscript has been judged scientifically suitable for publication and will be formally accepted for publication once it meets all outstanding technical requirements.

Kind regards,

Palash Chandra Banik, MPhil

Academic Editor

PLOS ONE

Additional Editor Comments (optional):

Reviewers' comments:

Reviewer's Responses to Questions

**Comments to the Author**

1. Does the manuscript provide a valid rationale for the proposed study, with clearly identified and justified research questions?

Reviewer #1: Yes

Reviewer #2: Yes

2. Is the protocol technically sound and planned in a manner that will lead to a meaningful outcome and allow testing the stated hypotheses?

Reviewer #1: Yes

Reviewer #2: Yes

3. Is the methodology feasible and described in sufficient detail to allow the work to be replicable?

Reviewer #1: Yes

Reviewer #2: Yes

4. Have the authors described where all data underlying the findings will be made available when the study is complete?

Reviewer #1: Yes

Reviewer #2: Yes

5. Is the manuscript presented in an intelligible fashion and written in standard English?

Reviewer #1: Yes

Reviewer #2: Yes

6. Review Comments to the Author

You may also provide optional suggestions and comments to authors that they might find helpful in planning their study.

Reviewer #1: After the revision of the protocol for the Scoping Review, the article now clearly presents a well-structured plan for the review itself. The only comment that could be considered is reference 24. This could be replaced with the reference "Best practice guidance and reporting items for the development of scoping review protocols" by Micah D.J. Peters et al., as it is a more recent source.

Reviewer #2: The revised manuscript offers more context and provides a valuable contribution to science. I am looking forward to the actual review. Still not very fond of literature review protocols being published but the Introduction section in itself brings an interesting point of view.

7. PLOS authors have the option to publish the peer review history of their article (what does this mean?). If published, this will include your full peer review and any attached files.

Reviewer #1: No

Reviewer #2: No

---

## [Editor Report · Acceptance letter]

8 Aug 2024

PONE-D-24-03451R1 

PLOS ONE

Dear Dr. Leal de Araújo, 

I'm pleased to inform you that your manuscript has been deemed suitable for publication in PLOS ONE. Congratulations! Your manuscript is now being handed over to our production team.

Kind regards, 

on behalf of

Dr. Palash Chandra Banik 

Academic Editor

PLOS ONE